# Cognitive Architectures for Language Agents

**Theodore R. Sumers**\*   **Shunyu Yao**\*   **Karthik Narasimhan**   **Thomas L. Griffiths**
*Princeton University*
*{sumers, shunyuy, karthikn, tomg}@princeton.edu*

**Reviewed on OpenReview:** *https://openreview.net/forum?id=1i6ZCvflQJ*

## Abstract

Recent efforts have augmented large language models (LLMs) with external resources (e.g., the Internet) or internal control flows (e.g., prompt chaining) for tasks requiring grounding or reasoning, leading to a new class of *language agents*. While these agents have achieved substantial empirical success, we lack a framework to organize existing agents and plan future developments. In this paper, we draw on the rich history of cognitive science and symbolic artificial intelligence to propose Cognitive Architectures for Language Agents (CoALA). CoALA describes a language agent with modular memory components, a structured action space to interact with internal memory and external environments, and a generalized decision-making process to choose actions. We use CoALA to *retrospectively* survey and organize a large body of recent work, and *prospectively* identify actionable directions towards more capable agents. Taken together, CoALA contextualizes today's language agents within the broader history of AI and outlines a path towards language-based general intelligence.

## 1 Introduction

*Language agents* (Weng, 2023; Wang et al., 2023b; Xi et al., 2023; Yao and Narasimhan, 2023) are an emerging class of artifical intelligence (AI) systems that use large language models (LLMs; Vaswani et al., 2017; Brown et al., 2020; Devlin et al., 2019; OpenAI, 2023a) to interact with the world. They apply the latest advances in LLMs to the existing field of agent design (Russell and Norvig, 2013). Intriguingly, this synthesis offers benefits for both fields. On one hand, LLMs possess limited knowledge and reasoning capabilities. Language agents mitigate these issues by connecting LLMs to internal memory and environments, grounding them to existing knowledge or external observations. On the other hand, traditional agents often require handcrafted rules (Wilkins, 2014) or reinforcement learning (Sutton and Barto, 2018), making generalization to new environments challenging (Lake et al., 2016). Language agents leverage commonsense priors present in LLMs to adapt to novel tasks, reducing the dependence on human annotation or trial-and-error learning.

While the earliest agents used LLMs to directly select or generate actions (Figure 1B; Ahn et al., 2022; Huang et al., 2022b), more recent agents additionally use them to reason (Yao et al., 2022b), plan (Hao et al., 2023; Yao et al., 2023), and manage long-term memory (Park et al., 2023; Wang et al., 2023a) to improve decision-making. This latest generation of *cognitive* language agents use remarkably sophisticated internal processes (Figure 1C). Today, however, individual works use custom terminology to describe these processes (such as 'tool use', 'grounding', 'actions'), making it difficult to compare different agents, understand how they are evolving over time, or build new agents with clean and consistent abstractions.

In order to establish a conceptual framework organizing these efforts, we draw parallels with two ideas from the history of computing and artificial intelligence (AI): *production systems* and *cognitive architectures*. Production systems generate a set of outcomes by iteratively applying rules (Newell and Simon, 1972). They originated as string manipulation systems – an analog of the problem that LLMs solve – and were subsequently adopted by the AI community to define systems capable of complex, hierarchically structured

---

\*Equal contribution, order decided by coin flip. Each person reserves the right to list their name first. A CoALA-based repo of recent work on language agents: `https://github.com/ysymyth/awesome-language-agents`.

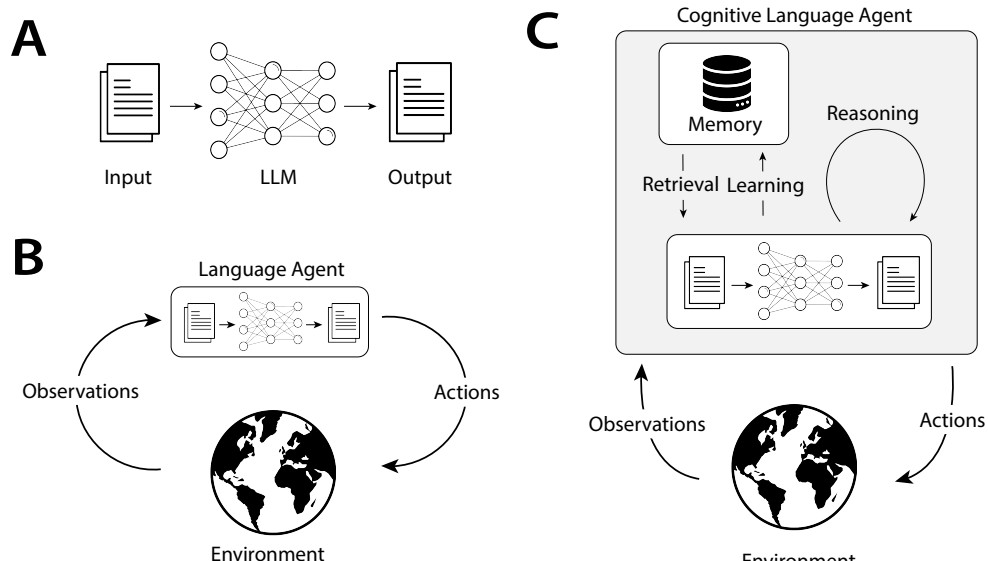

Figure 1: Different uses of large language models (LLMs). **A**: In natural language processing (NLP), an LLM takes text as input and outputs text. **B**: *Language agents* (Ahn et al., 2022; Huang et al., 2022c) place the LLM in a direct feedback loop with the external environment by transforming observations into text and using the LLM to choose actions. **C**: *Cognitive* language agents (Yao et al., 2022b; Shinn et al., 2023; Wang et al., 2023a) additionally use the LLM to manage the agent's internal state via processes such as learning and reasoning. In this work, we propose a blueprint to structure such agents.

behaviors (Newell et al., 1989). To do so, they were incorporated into cognitive architectures that specified control flow for selecting, applying, and even generating new productions (Laird et al., 1987; Laird, 2022; Kotseruba and Tsotsos, 2020). We suggest a meaningful analogy between production systems and LLMs: just as productions indicate possible ways to modify strings, LLMs define a distribution over changes or additions to text. This further suggests that controls from cognitive architectures used with production systems might be equally applicable to transform LLMs into language agents.

Thus, we propose **Co**gnitive **A**rchitectures for **L**anguage **A**gents (CoALA), a conceptual framework to characterize and design general purpose language agents. CoALA organizes agents along three key dimensions: their *information storage* (divided into working and long-term memories); their *action space* (divided into internal and external actions); and their *decision-making procedure* (which is structured as an interactive loop with planning and execution). Through these three concepts (memory, action, and decision-making), we show CoALA can neatly express a large body of existing agents and identify underexplored directions to develop new ones. Notably, while several recent papers propose conceptual architectures for general intelligence (LeCun, 2022; McClelland et al., 2019) or empirically survey language models and agents (Mialon et al., 2023; Weng, 2023; Wang et al., 2023b), this paper combines elements of both: we propose a theoretical framework *and* use it to organize diverse empirical work. This grounds our theory to existing practices and allows us to identify both short-term and long-term directions for future work.

The plan for the rest of the paper is as follows. We first introduce production systems and cognitive architectures (Section 2) and show how these recent developments in LLMs and language agents recapitulate these historical ideas (Section 3). Motivated by these parallels, Section 4 introduces the CoALA framework and uses it to survey existing language agents. Section 5 provides a deeper case study of several prominent agents. Section 6 suggests actionable steps to construct future language agents, while Section 7 highlights open questions in the broader arc of cognitive science and AI. Finally, Section 8 concludes. Readers interested in applied agent design may prioritize Sections 4-6.

## 2 Background: From Strings to Symbolic AGI

We first introduce production systems and cognitive architectures, providing a historical perspective on cognitive science and artificial intelligence: beginning with theories of logic and computation (Post, 1943), and ending with attempts to build symbolic artificial general intelligence (Newell et al., 1989). We then briefly introduce language models and language agents. Section 3 will connect these ideas, drawing parallels between production systems and language models.

### 2.1 Production systems for string manipulation

In the first half of the twentieth century, a significant line of intellectual work led to the reduction of mathematics (Whitehead and Russell, 1997) and computation (Church, 1932; Turing et al., 1936) to symbolic manipulation. Production systems are one such formalism. Intuitively, production systems consist of a set of rules, each specifying a precondition and an action. When the precondition is met, the action can be taken. The idea originates in efforts to characterize the limits of computation. Post (1943) proposed thinking about arbitrary logical systems in these terms, where formulas are expressed as strings and the conclusions they license are identified by production rules (as one string "produces" another). This formulation was subsequently shown to be equivalent to a simpler string rewriting system. In such a system, we specify rules of the form

$$X\,Y\,Z \to X\,W\,Z$$

indicating that the string $XYZ$ can be rewritten to the string $XWZ$. String rewriting plays a significant role in the theory of formal languages, in the form of Chomsky's phrase structure grammar (Chomsky, 1956).

### 2.2 Control flow: From strings to algorithms

By itself, a production system simply characterizes the set of strings that can be generated from a starting point. However, they can be used to specify algorithms if we impose *control flow* to determine which productions are executed. For example, Markov algorithms are production systems with a priority ordering (Markov, 1954). The following algorithm implements division-with-remainder by converting a number written as strokes | into the form $Q * R$, where $Q$ is the quotient of division by 5 and $R$ is the remainder:

$$
\begin{aligned}
*||||| &\to |* \\
* &\overset{\bullet}{\to} * \\
&\to *
\end{aligned}
$$

where the priority order runs from top to bottom, productions are applied to the first substring matching their preconditions when moving from left to right (including the empty substring, in the last production), and $\overset{\bullet}{\to}$ indicates the algorithm halts after executing the rule. The first rule effectively "subtracts" five if possible; the second handles the termination condition when no more subtraction is possible; and the third handles the empty substring input case. For example, given the input 11, this would yield the sequence of productions $*||||||||||| \to |*||||| \to ||*| \overset{\bullet}{\to} ||*|$ which is interpreted as 2 remainder 1. Simple productions can result in complex behavior – Markov algorithms can be shown to be Turing complete.

### 2.3 Cognitive architectures: From algorithms to agents

Production systems were popularized in the AI community by Allen Newell, who was looking for a formalism to capture human problem solving (Newell, 1967; Newell and Simon, 1972). Productions were generalized beyond string rewriting to logical operations: *preconditions* that could be checked against the agent's goals and world state, and *actions* that should be taken if the preconditions were satisfied. In their landmark book *Human Problem Solving* (Newell and Simon, 1972), Allen Newell and Herbert Simon gave the example of a

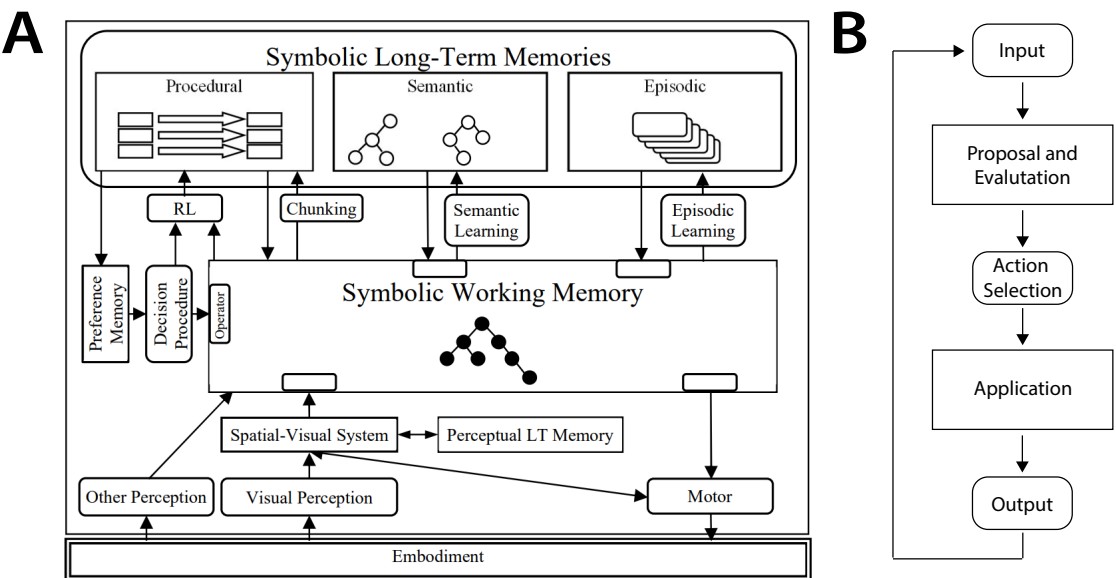

Figure 2: Cognitive architectures augment a production system with sensory groundings, long-term memory, and a decision procedure for selecting actions. **A**: The Soar architecture, reproduced with permission from Laird (2022). **B**: Soar's decision procedure uses productions to select and implement actions. These actions may be *internal* (such as modifying the agent's memory) or *external* (such as a motor command).

simple production system implementing a thermostat agent:

$$(\text{temperature} > 70°) \wedge (\text{temperature} < 72°) \quad \rightarrow \quad \text{stop}$$
$$\text{temperature} < 32° \quad \rightarrow \quad \text{call for repairs; turn on electric heater}$$
$$(\text{temperature} < 70°) \wedge (\text{furnace off}) \quad \rightarrow \quad \text{turn on furnace}$$
$$(\text{temperature} > 72°) \wedge (\text{furnace on}) \quad \rightarrow \quad \text{turn off furnace}$$

Following this work, production systems were adopted by the AI community. The resulting agents contained large production systems connected to external sensors, actuators, and knowledge bases – requiring correspondingly sophisticated control flow. AI researchers defined "cognitive architectures" that mimicked human cognition – explicitly instantiating processes such as perception, memory, and planning (Adams et al., 2012) to achieve flexible, rational, real-time behaviors (Sun, 2004; Newell, 1980; 1992; Anderson and Lebiere, 2003). This led to applications from psychological modeling to robotics, with hundreds of architectures and thousands of publications (see Kotseruba and Tsotsos (2020) for a recent survey).

A canonical example is the Soar architecture (Fig. 2A). Soar stores productions in long-term memory and executes them based on how well their preconditions match working memory (Fig. 2B). These productions specify actions that modify the contents of working and long-term memory. We next provide a brief overview of Soar and refer readers to Laird (2022; 2019) for deeper introductions.

**Memory.** Building on psychological theories, Soar uses several types of memory to track the agent's state (Atkinson and Shiffrin, 1968). *Working memory* (Baddeley and Hitch, 1974) reflects the agent's current circumstances: it stores the agent's recent perceptual input, goals, and results from intermediate, internal reasoning. *Long term memory* is divided into three distinct types. *Procedural* memory stores the production system itself: the set of rules that can be applied to working memory to determine the agent's behavior. *Semantic* memory stores facts about the world (Lindes and Laird, 2016), while *episodic* memory stores sequences of the agent's past behaviors (Nuxoll and Laird, 2007).

**Grounding.** Soar can be instantiated in simulations (Tambe et al., 1995; Jones et al., 1999) or real-world robotic systems (Laird et al., 2012). In embodied contexts, a variety of sensors stream perceptual input into

working memory, where it is available for decision-making. Soar agents can also be equipped with actuators, allowing for physical actions and interactive learning via language (Mohan et al., 2012; Mohan and Laird, 2014; Kirk and Laird, 2014).

**Decision making.** Soar implements a decision loop that evaluates productions and applies the one that matches best (Fig. 2B). Productions are stored in long-term procedural memory. During each decision cycle, their preconditions are checked against the agent's working memory. In the *proposal and evaluation* phase, a set of productions is used to generate and rank a candidate set of possible actions.[*] The best action is then chosen.[†] Another set of productions is then used to implement the action – for example, modifying the contents of working memory or issuing a motor command.

**Learning.** Soar supports multiple modes of learning. First, new information can be stored directly in long-term memory: facts can be written to semantic memory, while experiences can be written to episodic memory (Derbinsky et al., 2012). This information can later be retrieved back into working memory when needed for decision-making. Second, behaviors can be modified. Reinforcement learning (Sutton and Barto, 2018) can be used to up-weight productions that have yielded good outcomes, allowing the agent to learn from experience (Nason and Laird, 2005). Most remarkably, Soar is also capable of writing new productions into its procedural memory (Laird et al., 1986) – effectively updating its source code.

Cognitive architectures were used broadly across psychology and computer science, with applications including robotics (Laird et al., 2012), military simulations (Jones et al., 1999; Tambe et al., 1995), and intelligent tutoring (Koedinger et al., 1997). Yet they have become less popular in the AI community over the last few decades. This decrease in popularity reflects two of the challenges involved in such systems: they are limited to domains that can be described by logical predicates and require many pre-specified rules to function.

Intriguingly, LLMs appear well-posed to meet these challenges. First, they operate over arbitrary text, making them more flexible than logic-based systems. Second, rather than requiring the user to specify productions, they learn a distribution over productions via pre-training on an internet corpus. Recognizing this, researchers have begun to use LLMs within cognitive architectures, leveraging their implicit world knowledge (Wray et al., 2021) to augment traditional symbolic approaches (Kirk et al., 2023; Romero et al., 2023). Here, we instead import principles from cognitive architecture to guide the design of LLM-based agents.

## 2.4 Language models and agents

Language modeling is a decades-old endeavor in the NLP and AI communities, aiming to develop systems that can generate text given some context (Jurafsky, 2000). Formally, language models learn a distribution $P(w_i|w_{<i})$, where each $w$ is an individual token (word). This model can then generate text by sampling from the distribution, one token at a time. At its core, a language model is a probabilistic input-output system, since there are inherently several ways to continue a text (e.g., "I went to the" $\rightarrow$ "market" | "beach" | ...). While earlier attempts at modeling language (e.g., n-grams) faced challenges in generalization and scaling, there has been a recent resurgence of the area due to the rise of Transformer-based (Vaswani et al., 2017) LLMs with a large number (billions) of parameters (e.g., GPT-4; OpenAI, 2023a) and smart tokenization schemes. Modern LLMs are trained on enormous amounts of data, which helps them accumulate knowledge from a large number of input-output combinations and successfully generate human-like text (Andreas, 2022).

Unexpectedly, training these models on internet-scale text also made them useful for many tasks beyond generating text, such as writing code (Li et al., 2022b; Rozière et al., 2023; Li et al., 2023c), modeling proteins (Meier et al., 2021), and acting in interactive environments (Yao et al., 2022b; Nakano et al., 2021). The latter has led to the rise of "language agents" – systems that use LLMs as a core computation unit to reason, plan, and act – with applications in areas such as robotics (Ahn et al., 2022), manufacturing (Xia et al., 2023), web manipulation (Yao et al., 2022a; Deng et al., 2023), puzzle solving (Yao et al., 2023; Hao et al., 2023) and interactive code generation (Yang et al., 2023). The combination of language understanding

---

[*]In more detail, Soar divides productions into two types: "operators," which we refer to as actions, and "rules" which are used to propose, evaluate, and execute operators.

[†]If no actions are valid, or multiple actions tie, then an *impasse* occurs. Soar creates a subgoal to resolve the impasse, resulting in hierarchical task decomposition. We refer the reader to Laird (2022) for a more detailed discussion.

and decision-making capabilities is an exciting and emerging direction that promises to bring these agents closer to human-like intelligence.

## 3 Connections between Language Models and Production Systems

Based on their common origins in processing strings, there is a natural analogy between production systems and language models. We develop this analogy, then show that prompting methods recapitulate the algorithms and agents based on production systems. The correspondence between production systems and language models motivates our use of cognitive architectures to build language agents, which we introduce in Section 4.

### 3.1 Language models as probabilistic production systems

In their original instantiation, production systems specified the set of strings that could be generated from a starting point, breaking this process down into a series of string rewriting operations. Language models also define a possible set of expansions or modifications of a string – the prompt provided to the model.[‡]

For example, we can formulate the problem of completing a piece of text as a production. If $X$ is the prompt and $Y$ the continuation, then we can write this as the production $X \rightarrow X\,Y$.[§] We might want to allow multiple possible continuations, in which case we have $X \rightarrow X\,Y_i$ for some set of $Y_i$. LLMs assign a *probability* to each of these completions. Viewed from this perspective, the LLM defines a probability distribution over *which productions to select* when presented with input $X$, yielding a distribution $P(Y_i|X)$ over possible completions (Dohan et al., 2022). LLMs can thus be viewed as probabilistic production systems that sample a possible completion each time they are called, e.g., $X \rightsquigarrow X\,Y$.

This probabilistic form offers both advantages and disadvantages compared to traditional production systems. The primary disadvantage of LLMs is their inherent opaqueness: while production systems are defined by discrete and human-legible rules, LLMs consist of billions of uninterpretable parameters. This opaqueness – coupled with inherent randomness from their probabilistic formulation – makes it challenging to analyze or control their behaviors (Romero et al., 2023; Valmeekam et al., 2022). Nonetheless, their scale and pre-training provide massive advantages over traditional production systems. LLMs pre-trained on large-scale internet data learn a remarkably effective prior over string completions, allowing them to solve a wide range of tasks out of the box (Huang et al., 2022b).

### 3.2 Prompt engineering as control flow

The weights of an LLM define a prioritization over output strings (completions), conditioned by the input string (the prompt). The resulting distribution can be interpreted as a task-specific prioritization of productions – in other words, a simple control flow. Tasks such as question answering can be formulated directly as an input string (the question), yielding conditional distributions over completions (possible answers).

Early work on few-shot learning (Brown et al., 2020) and prompt engineering (Wei et al., 2022b; Kojima et al., 2022; Xu et al., 2023c) found that the LLM could be further biased towards high-quality productions by pre-processing the input string. These simple manipulations – typically concatenating additional text to the input – can themselves be seen as productions, meaning that these methods define a sequence of productions (Table 1). Later work extended these approaches to dynamic, context-sensitive prompts: for example, selecting few-shot examples that are maximally relevant to the input (Liu et al., 2021) or populating a template with external observations from video (Zeng et al., 2022) or databases (Lewis et al., 2020). For a survey of such prompting techniques, see Liu et al. (2023d).

Subsequent work used the LLM itself as a pre-processing step, eliciting targeted reasoning to foreground a particular aspect of the problem (Bai et al., 2022; Jin et al., 2022; Ganguli et al., 2023; Madaan et al., 2023; Saunders et al., 2022; Kim et al., 2023; Kirk et al., 2023) or generate intermediate reasoning steps (Tafjord

---

[‡]In this work, we focus on autoregressive LLMs which are typically used for language agents. However, bidirectional LLMs such as BERT (Devlin et al., 2019) can be seen in a similar light: they define a distribution over *in-filling* productions.

[§]Alternatively, we can treat the prompt as input and take the output of the LLM as the next state, represented by the production $X \rightarrow Y$ – a more literal form of rewriting.

| Prompting Method | Production Sequence |
|---|---|
| Zero-shot | $Q \overset{\text{LLM}}{\rightsquigarrow} Q\ A$ |
| Few-shot | $Q \longrightarrow Q_1\ A_1\ Q_2\ A_2\ Q \overset{\text{LLM}}{\rightsquigarrow} Q_1\ A_1\ Q_2\ A_2\ Q\ A$ |
| Retrieval Augmented Generation | $Q \overset{\text{Wiki}}{\longrightarrow} Q\ O \overset{\text{LLM}}{\rightsquigarrow} Q\ O\ A$ |
| Socratic Models | $Q \overset{\text{VLM}}{\rightsquigarrow} Q\ O \overset{\text{LLM}}{\rightsquigarrow} Q\ O\ A$ |
| Self-Critique | $Q \overset{\text{LLM}}{\rightsquigarrow} Q\ A \overset{\text{LLM}}{\rightsquigarrow} Q\ A\ C \overset{\text{LLM}}{\rightsquigarrow} Q\ A\ C\ A$ |

Table 1: Conceptual diagram illustrating how prompting methods manipulate the input string before generating completions. $Q$ = question, $A$ = answer, $O$ = observation, $C$ = critique, and $\rightsquigarrow$ denotes sampling from a stochastic production. These pre-processing manipulations – which can employ other models such as vision-language models (VLMs), or even the LLM itself – can be seen as productions. Prompting methods thus define a *sequence* of productions.

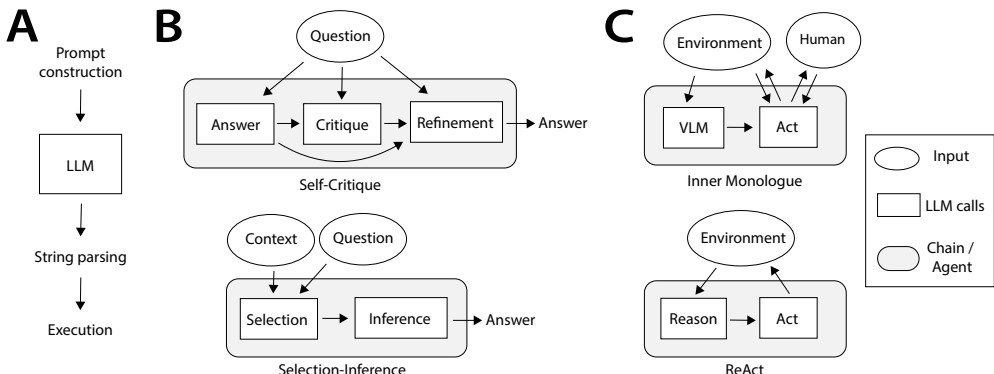

Figure 3: From language models to language agents. **A**: Basic structure of an LLM call. Prompt construction selects a template and populates it with variables from working memory. After calling the LLM, the string output is parsed into an action space and executed. An LLM call may result in one or more actions – for example, returning an answer, calling a function, or issuing motor commands. **B**: *Prompt chaining* techniques such as Self-Critique (Wang et al., 2022b) or Selection-Inference (Creswell et al., 2023) use a pre-defined sequence of LLM calls to generate an output. **C**: *Language agents* such as Inner Monologue (Huang et al., 2022c) and ReAct (Yao et al., 2022b) instead use an interactive feedback loop with the external environment. Vision-language models (VLMs) can be used to translate perceptual data into text for the LLM to process.

et al., 2021; Creswell et al., 2023; Yao et al., 2023) before returning an answer. *Chaining* multiple calls to an LLM (Wu et al., 2022a;b; Dohan et al., 2022) allows for increasingly complicated algorithms (Fig. 3).

### 3.3   Towards cognitive language agents

*Language agents* move beyond pre-defined prompt chains and instead place the LLM in a feedback loop with the external environment (Fig. 1B). These approaches first transform multimodal input into text and pass it to the LLM. The LLM's output is then parsed and used to determine an external action (Fig. 3C). Early agents interfaced the LLM directly with the external environment, using it to produce high-level instructions based on the agent's state (Ahn et al., 2022; Huang et al., 2022c; Dasgupta et al., 2022). Later work developed more sophisticated language agents that use the LLM to perform intermediate reasoning before selecting an action (Yao et al., 2022b). The most recent agents incorporate sophisticated learning strategies such as reflecting on episodic memory to generate new semantic inferences (Shinn et al., 2023) or modifying their

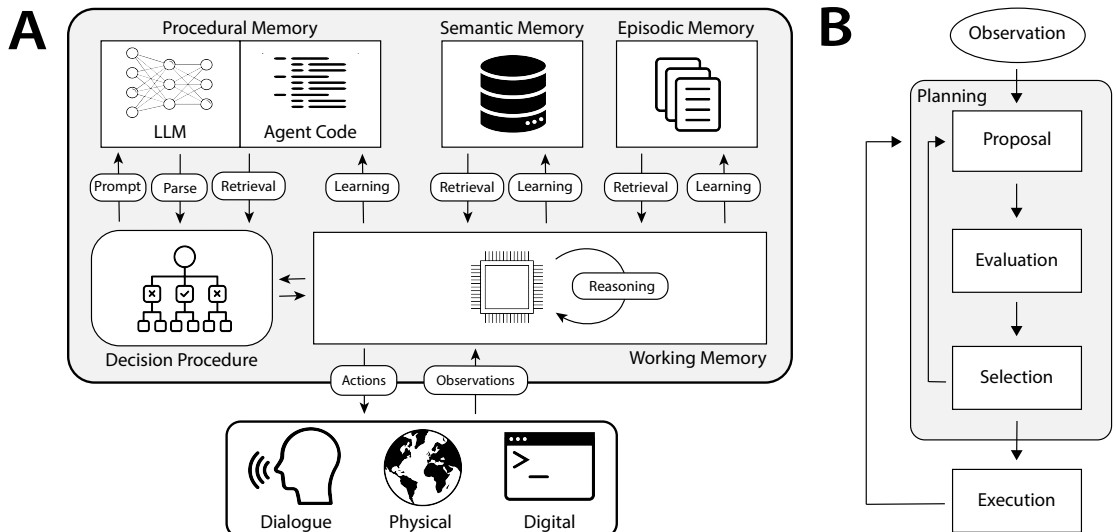

Figure 4: Cognitive architectures for language agents (CoALA). **A**: CoALA defines a set of interacting modules and processes. The **decision procedure** executes the agent's source code. This source code consists of procedures to interact with the LLM (prompt templates and parsers), internal memories (retrieval and learning), and the external environment (grounding). **B**: Temporally, the agent's decision procedure executes a **decision cycle** in a loop with the external environment. During each cycle, the agent uses **retrieval** and **reasoning** to plan by proposing and evaluating candidate **learning** or **grounding** actions. The best action is then selected and executed. An observation may be made, and the cycle begins again.

program code to generate procedural knowledge (Wang et al., 2023a), using their previous experience to adapt their future behaviors.

These *cognitive* language agents employ nontrivial LLM-based reasoning and learning (Fig. 1C). Just as cognitive architectures were used to structure production systems' interactions with agents' internal state and external environments, we suggest that they can help design LLM-based cognitive agents. In the remainder of the paper, we use this perspective to organize existing approaches and highlight promising extensions.

## 4    Cognitive Architectures for Language Agents (CoALA): A Conceptual Framework

We present Cognitive Architectures for Language Agents (CoALA) as a framework to organize existing language agents and guide the development of new ones. CoALA positions the LLM as the core component of a larger cognitive architecture (Figure 4). Under CoALA, a language agent stores information in **memory** modules (Section 4.1), and acts in an action space structured into external and internal parts (Figure 5):

- **External actions** interact with external environments (e.g., control a robot, communicate with a human, navigate a website) through **grounding** (Section 4.2).

- **Internal actions** interact with internal memories. Depending on which memory gets accessed and whether the access is read or write, internal actions can be further decomposed into three kinds: **retrieval** (read from long-term memory; Section 4.3), **reasoning** (update the short-term working memory with LLM; Section 4.4), and **learning** (write to long-term memory; Section 4.5).

Language agents choose actions via **decision-making**, which follows a repeated cycle (Section 4.6, Figure 4B). In each cycle, the agent can use reasoning and retrieval actions to plan. This planning subprocess selects a grounding or learning action, which is executed to affect the outside world or the agent's long-term memory. CoALA's decision cycle is analogous to a program's "main" *procedure* (a *method* without return values, as

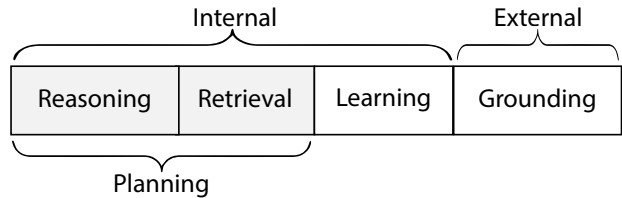

Figure 5: Agents' action spaces can be divided into **internal** memory accesses and **external** interactions with the world. **Reasoning** and **retrieval** actions are used to support planning.

opposed to *functions*) that runs in loops continuously, accepting new perceptual input and calling various action *procedures* in response.

CoALA (Figure 4) is inspired by the decades of research in cognitive architectures (Section 2.3), leveraging key concepts such as memory, grounding, learning, and decision-making. Yet the incorporation of an LLM leads to the addition of "reasoning" actions, which can flexibly produce new knowledge and heuristics for various purposes – replacing hand-written rules in traditional cognitive architectures. It also makes text the *de facto* internal representation, streamlining agents' memory modules. Finally, recent advances in vision-language models (VLMs; Alayrac et al., 2022) can simplify grounding by providing a straightforward translation of perceptual data into text (Zeng et al., 2022).

The rest of this section details key concepts in CoALA: memory, actions (grounding, reasoning, retrieval, and learning), and decision-making. For each concept, we use existing language agents (or relevant NLP/RL methods) as examples – or note gaps in the literature for future directions.

## 4.1 Memory

Language models are *stateless*: they do not persist information across calls. In contrast, language agents may store and maintain information internally for multi-step interaction with the world. Under the CoALA framework, language agents explicitly organize information (mainly textural, but other modalities also allowed) into multiple memory modules, each containing a different form of information. These include short-term working memory and several long-term memories: episodic, semantic, and procedural.

**Working memory**. Working memory maintains active and readily available information as symbolic variables for the current decision cycle (Section 4.6). This includes perceptual inputs, active knowledge (generated by reasoning or retrieved from long-term memory), and other core information carried over from the previous decision cycle (e.g., agent's active goals). Previous methods encourage the LLM to generate intermediate reasoning (Wei et al., 2022b; Nye et al., 2021), using the LLM's own context as a form of working memory. CoALA's notion of working memory is more general: it is a data structure that persists across LLM calls. On each LLM call, the LLM input is synthesized from a subset of working memory (e.g., a prompt template and relevant variables). The LLM output is then parsed back into other variables (e.g., an action name and arguments) which are stored back in working memory and used to execute the corresponding action (Figure 3A). Besides the LLM, the working memory also interacts with long-term memories and grounding interfaces. It thus serves as the central hub connecting different components of a language agent.

**Episodic memory**. Episodic memory stores experience from earlier decision cycles. This can consist of training input-output pairs (Rubin et al., 2021), history event flows (Weston et al., 2014; Park et al., 2023), game trajectories from previous episodes (Yao et al., 2020; Tuyls et al., 2022), or other representations of the agent's experiences. During the planning stage of a decision cycle, these episodes may be retrieved into working memory to support reasoning. An agent can also write new experiences from working to episodic memory as a form of learning (Section 4.5).

**Semantic memory**. Semantic memory stores an agent's knowledge about the world and itself. Traditional NLP or RL approaches that leverage retrieval for reasoning or decision-making initialize semantic memory from an external database for knowledge support. For example, retrieval-augmented methods in NLP (Lewis et al., 2020; Borgeaud et al., 2022; Chen et al., 2017) can be viewed as retrieving from a semantic memory of

unstructured text (e.g., Wikipedia). In RL, "reading to learn" approaches (Branavan et al., 2012; Narasimhan et al., 2018; Hanjie et al., 2021; Zhong et al., 2021) leverage game manuals and facts as a semantic memory to affect the policy. While these examples essentially employ a fixed, read-only semantic memory, language agents may also write new knowledge obtained from LLM reasoning into semantic memory as a form of learning (Section 4.5) to incrementally build up world knowledge from experience.

**Procedural memory**. Language agents contain two forms of procedural memory: *implicit* knowledge stored in the LLM weights, and *explicit* knowledge written in the agent's code. The agent's code can be further divided into two types: procedures that implement actions (reasoning, retrieval, grounding, and learning procedures), and procedures that implement decision-making itself (Section 4.6). During a decision cycle, the LLM can be accessed via reasoning actions, and various code-based procedures can be retrieved and executed. Unlike episodic or semantic memory that may be initially empty or even absent, procedural memory must be initialized by the designer with proper code to bootstrap the agent. Finally, while learning new actions by writing to procedural memory is possible (Section 4.5), it is significantly riskier than writing to episodic or semantic memory, as it can easily introduce bugs or allow an agent to subvert its designers' intentions.

## 4.2 Grounding actions

Grounding procedures execute external actions and process environmental feedback into working memory as text. This effectively simplifies the agent's interaction with the outside world as a "text game" with textual observations and actions. We categorize three kinds of external environments:

**Physical environments**. Physical embodiment is the oldest instantiation envisioned for AI agents (Nilsson, 1984). It involves processing perceptual inputs (visual, audio, tactile) into textual observations (e.g., via pre-trained captioning models), and affecting the physical environments via robotic planners that take language-based commands. Recent advances in LLMs have led to numerous robotic projects (Ahn et al., 2022; Liang et al., 2023a; Singh et al., 2023; Palo et al., 2023; Ren et al., 2023) that leverage LLMs as a "brain" for robots to generate actions or plans in the physical world. For perceptual input, vision-language models are typically used to convert images to text (Alayrac et al., 2022; Sumers et al., 2023) providing additional context for the LLM (Driess et al., 2023; Huang et al., 2023; Brohan et al., 2022; 2023).

**Dialogue with humans or other agents**. Classic linguistic interactions allow the agent to accept instructions (Winograd, 1972; Tellex et al., 2011; Chen and Mooney, 2011; Bisk et al., 2016) or learn from people (Nguyen et al., 2021; Sumers et al., 2022; 2021; Wang et al., 2016). Agents capable of *generating* language may ask for help (Ren et al., 2023; Nguyen et al., 2022b; 2019; Nguyen and Daumé III, 2019) or clarification (Biyik and Palan, 2019; Sadigh et al., 2017; Padmakumar et al., 2022; Thomason et al., 2020; Narayan-Chen et al., 2019) – or entertain or emotionally help people (Zhang et al., 2020; Zhou et al., 2018; Pataranutaporn et al., 2021; Hasan et al., 2023; Ma et al., 2023). Recent work also investigates interaction among multiple language agents for social simulation (Park et al., 2023; Jinxin et al., 2023; Gao et al., 2023), debate (Chan et al., 2023; Liang et al., 2023b; Du et al., 2023), improved safety (Irving et al., 2018), or collaborative task solving (Qian et al., 2023; Wu et al., 2023; Hong et al., 2023a; Dong et al., 2023).

**Digital environments**. This includes interacting with games (Hausknecht et al., 2020; Côté et al., 2019; Shridhar et al., 2020; Wang et al., 2022a; Liu et al., 2023e), APIs (Schick et al., 2023; Yao et al., 2022b; Parisi et al., 2022; Tang et al., 2023b), and websites (Shi et al., 2017; Nakano et al., 2021; Yao et al., 2022a; Zhou et al., 2023b; Gur et al., 2023; Deng et al., 2023) as well as general code execution (Yang et al., 2023; Le et al., 2022; Ni et al., 2023). Such digital grounding is cheaper and faster than physical or human interaction. It is thus a convenient testbed for language agents and has been studied with increasing intensity in recent years. In particular, for NLP tasks that require augmentation of external knowledge or computation, stateless digital APIs (e.g., search, calculator, translator) are often packaged as "**tools**" (Parisi et al., 2022; Schick et al., 2023; Xu et al., 2023a; Tang et al., 2023b; Qin et al., 2023), which can be viewed as special "single-use" digital environments.

### 4.3 Retrieval actions

In CoALA, a retrieval procedure (Li et al., 2022a; Gu et al., 2018) reads information from long-term memories into working memory. Depending on the information and memory type, it could be implemented in various ways, e.g., rule-based, sparse, or dense retrieval. For example, Voyager (Wang et al., 2023a) loads code-based skills from a skill library via dense retrieval to interact with the Minecraft world – effectively retrieving grounding procedures from a procedural memory. Generative Agents (Park et al., 2023) retrieves relevant events from episodic memory via a combination of recency (rule-based), importance (reasoning-based), and relevance (embedding-based) scores. DocPrompting (Zhou et al., 2022a) proposes to leverage library documents to assist code generation, which can be seen as retrieving knowledge from semantic memory. While retrieval plays a key role in human decision-making (Zhou et al., 2023a; Zhao et al., 2022), adaptive and context-specific recall remains understudied in language agents. In Section 6, we suggest a principled integration of decision-making and retrieval as an important future direction.

### 4.4 Reasoning actions

Reasoning allows language agents to process the contents of working memory to generate new information. Unlike retrieval (which reads from long-term memory into working memory), reasoning reads from *and* writes to working memory. This allows the agent to summarize and distill insights about the most recent observation (Yao et al., 2022b; Peng et al., 2023), the most recent trajectory (Shinn et al., 2023), or information retrieved from long-term memory (Park et al., 2023). Reasoning can be used to support learning (by writing the results into long-term memory) or decision-making (by using the results as additional context for subsequent LLM calls).

### 4.5 Learning actions

Learning occurs by writing information to long-term memory, which includes a spectrum of diverse procedures.

**Updating episodic memory with experience.** It is common practice for RL agents to store episodic trajectories to update a parametric policy (Blundell et al., 2016; Pritzel et al., 2017) or establish a non-parametric policy (Ecoffet et al., 2019; Tuyls et al., 2022). For language agents, added experiences in episodic memory may be retrieved later as examples and bases for reasoning or decision-making (Weston et al., 2014; Rubin et al., 2021; Park et al., 2023).

**Updating semantic memory with knowledge.** Recent work (Shinn et al., 2023; Park et al., 2023) has applied LLMs to reason about raw experiences and store the resulting inferences in semantic memory. For example, Reflexion (Shinn et al., 2023) uses an LLM to reflect on failed episodes and stores the results (e.g., "there is no dishwasher in kitchen") as semantic knowledge to be attached to LLM context for solving later episodes. Finally, work in robotics (Chen et al., 2023a) uses vision-language models to build a semantic map of the environment, which can later be queried to execute instructions.

**Updating LLM parameters (procedural memory).** The LLM weights represent implicit procedural knowledge. These can be adjusted to an agent's domain by fine-tuning during the agent's lifetime. Such fine-tuning can be accomplished via supervised (Liu et al., 2023c; Zhang et al., 2023b) or imitation learning (Hussein et al., 2017), reinforcement learning (RL) from environment feedback (Sutton and Barto, 2018), human feedback (RLHF; Christiano et al., 2017; Ouyang et al., 2022; Nakano et al., 2021), or AI feedback (Bai et al., 2022; Liu et al., 2023f). Classic LLM self-improvement methods (Huang et al., 2022a; Zelikman et al., 2022) use an external measure such as consistency Wang et al. (2022b) to select generations to fine-tune on. In reinforcement learning settings, this can be extended to use environmental feedback instead: for example, XTX (Tuyls et al., 2022) periodically fine-tunes a small language model on high-scoring trajectories stored in episodic memory, which serves as a robust "exploitation" policy to reach exploration frontiers in the face of stochasity. Fine-tuning the agent's LLM is a costly form of learning; thus, present studies specify learning schedules. However, as training becomes more efficient – or if agents utilize smaller subtask-specific LLMs – it may be possible to allow language agents to autonomously determine when and how to fine-tune their LLMs.

**Updating agent code (procedural memory).** CoALA allows agents to update their source code, thus modifying the implementation of various procedures. These can be broken down as follows:

- **Updating reasoning** (e.g., prompt templates; Gao et al., 2020; Zhou et al., 2022b). For example, APE (Zhou et al., 2022b) infers prompt instructions from input-output examples, then uses these instructions as part of the LLM prompt to assist task solving. Such a prompt update can be seen as a form of learning to reason.

- **Updating grounding** (e.g., code-based skills; Liang et al., 2023a; Ellis et al., 2021; Wang et al., 2023a). For example, Voyager (Wang et al., 2023a) maintains a curriculum library. Notably, current methods are limited to creating new code skills to interact with external environments.

- **Updating retrieval**. To our knowledge, these learning options are not studied in recent language agents. Retrieval is usually considered a basic action designed with some fixed implementation (e.g., BM25 or dense retrieval), but research in query/document expansion (Nogueira et al., 2019; Wang et al., 2023c; Tang et al., 2023a) or retrieval distillion (Izacard et al., 2021) may be helpful for language agents to learn better retrieval procedures.

- **Updating learning or decision-making**. Finally, it is theoretically possible for CoALA agents to learn new procedures for learning or decision-making, thus providing significant adaptability. In general, however, updates to these procedures are risky both for the agent's functionality and alignment. At present, we are not aware of any language agents that implement this form of learning; we discuss such possibilities more in Section 6.

While RL agents usually fix one way of learning (e.g., Q-learning, PPO, or A3C) and learn by updating model parameters, language agents can select from a diversity of learning procedures. This allows them to learn rapidly by storing task-relevant language (cheaper and quicker than parameter updates), and leverage multiple forms of learning to compound their self-improvement (e.g., Generative Agents discussed in Section 5).

Finally, while our discussion has mostly focused on **adding** to memory, **modifying** and **deleting** (a case of "unlearning") are understudied in recent language agents. We address these areas more in Section 6.

## 4.6 Decision making

With various actions (grounding, learning, reasoning, retrieval) in the action space, how should a language agent choose which action to apply? This is handled by the decision-making procedure, which is effectively the top-level or "main" agent program. CoALA structures this top-level program into decision cycles (Figure 4B) which yield an external *grounding* action (Section 4.2) or internal *learning* action (Section 4.5). In each cycle, program code defines a sequence of reasoning and retrieval actions to propose and evaluate alternatives (**planning stage**), then executes the selected action (**execution stage**) – then the cycle loops again.

**Planning stage**. During planning, reasoning and retrieval can be flexibly applied to propose, evaluate, and select actions, and these sub-stages could interleave or iterate to build up multi-step simulations (Tamari et al., 2020) before taking an external action (Yao et al., 2023; Hao et al., 2023). It also enables agents to iteratively improve candidate solutions – for example, by using the LLM to simulate them, identifying defects, and proposing modifications that address those defects (Kirk et al., 2023; Shinn et al., 2023).

- **Proposal**. The proposal sub-stage generates one or more action candidates. The usual approach is to use **reasoning** (and optionally retrieval) to sample one (Huang et al., 2022c) or more (Chen et al., 2021; Wang et al., 2022b) external grounding actions from the LLM. For simple domains with limited actions, the proposal stage might simply include all actions (e.g., SayCan in Section 5). More sophisticated agents use if-else or while-if code structures (Wang et al., 2023a; Park et al., 2023); while agents deployed in well-defined domains may utilize structured simulators (Haslum et al., 2019) to generate plausible rollouts (Liu et al., 2023a; Dagan et al., 2023).

- **Evaluation**. If multiple actions are proposed, the evaluation sub-stage assigns a value to each. This may use heuristic rules, LLM (perplexity) values (Ahn et al., 2022), learned values (Yao et al.,

| | Long-term Memory¶ | External Grounding | Internal Actions | Decision Making |
|---|---|---|---|---|
| SayCan (Ahn et al., 2022) | - | physical | - | evaluate |
| ReAct (Yao et al., 2022b) | - | digital | reason | propose |
| Voyager (Wang et al., 2023a) | procedural | digital | reason/retrieve/learn | propose |
| Generative Agents (Park et al., 2023) | episodic/semantic | digital/agent | reason/retrieve/learn | propose |
| Tree of Thoughts (Yao et al., 2023) | - | digital‖ | reason | propose, evaluate, select |

Table 2: Some recent language agents cast into the CoALA framework.

2020), LLM reasoning (Yao et al., 2023; Hao et al., 2023), or some combination. Particularly, LLM reasoning can help evaluate actions by internally simulating their grounding feedback from the external world (Hao et al., 2023; Yang et al., 2023).

- **Selection**. Given a set of actions and their values, the selection step either selects one to execute or rejects them and loops back to the proposal step. Depending on the form of action values, selection may occur via argmax, softmax, or an alternative such as majority vote (Wang et al., 2022b).

**Execution**. The selected action is applied by executing the relevant procedures from the agent's source code. Depending on the agent implementation, this might be an external *grounding* action (e.g., an API call; Section 4.2) or an internal *learning* action (e.g., a write to episodic memory; Section 4.5). An observation can be made from the environment, providing feedback from the agent's action, and the cycle loops again.

Empirically, many early language agents simply use LLMs to propose an action (Schick et al., 2023), a sequence of actions (Huang et al., 2022b), or evaluate a fixed set of actions (Ahn et al., 2022) without intermediate reasoning or retrieval. Followup work (Yao et al., 2022b; Shinn et al., 2023; Xu et al., 2023b; Lin et al., 2023; Wang et al., 2023a; Park et al., 2023) has exploited intermediate reasoning and retrieval to analyze the situation, make and maintain action plans, refine the previous action given the environmental feedback, and leveraged a more complex procedure to propose a single action. Most recently, research has started to investigate more complex decision-making employing iterative proposal and evaluation to consider multiple actions. These procedures are modeled after classical planning algorithms: for example, Tree of Thoughts (Yao et al., 2023) and RAP (Hao et al., 2023) use LLMs to implement BFS/DFS and Monte Carlo Tree Search (MCTS; Browne et al., 2012) respectively. LLMs are used to generate proposals (i.e., to simulate rollouts conditioned on an action) and evaluate them (i.e., to value the outcome of the proposed action).

## 5   Case Studies

With variations and ablations of the memory modules, action space, and decision-making procedures, CoALA can express a wide spectrum of language agents. Table 2 lists some popular recent methods across diverse domains — from Minecraft to robotics, from pure reasoning to social simulacra. CoALA helps characterize their internal mechanisms and reveal their similarities and differences in a simple and structured way.

**SayCan** (Ahn et al., 2022) grounds a language model to robotic interactions in a kitchen to satisfy user commands (e.g., "I just worked out, can you bring me a drink and a snack to recover?"). Its long-term memory is procedural only (an LLM and a learned value function). The action space is external only – a fixed set of 551 grounding skills (e.g., "find the apple", "go to the table"), with no internal actions of reasoning, retrieval, or learning. During decision-making, SayCan evaluates each action using a combination of LLM and learned values, which balance a skill's usefulness and groundedness. SayCan therefore employs the LLM (in conjunction with the learned value function) as a single-step planner.

**ReAct** (Yao et al., 2022b) is a language agent grounded to various digital environments (e.g., Wikipedia API, text game, website). Like SayCan, it lacks semantic or episodic memory and therefore has no retrieval or learning actions. Its action space consists of (internal) reasoning and (external) grounding. Its decision cycle is fixed to use a single reasoning action to analyze the situation and (re)make action plans, then generates a

---

¶All agents contain some procedural memory (agent code and LLM weights), so here we only list writable procedural memory.
‖Special digital grounding with the only external action being submitting a final answer.

grounding action without evaluation or selection stages. ReAct can be considered the simplest language agent that leverages both internal and external actions, and is the initial work that demonstrates their synergizing effects: reasoning helps guide acting, while acting provides environmental feedback to support reasoning.

**Voyager** (Wang et al., 2023a) is a language agent grounded to the Minecraft API. Unlike SayCan, which grounds to perception via the learned value function, Voyager's grounding is text-only. It has a long-term procedural memory that stores a library of code-based grounding procedures a.k.a. skills (e.g., "combatZombie", "craftStoneSword"). This library is hierarchical: complex skills can use simpler skills as sub-procedures (e.g., "combatZombie" may call "craftStoneSword" if no sword is in inventory). Most impressively, its action space has all four kinds of actions: grounding, reasoning, retrieval, and learning (by adding new grounding procedures). During a decision cycle, Voyager first reasons to propose a new task objective if it is missing in the working memory, then reasons to propose a code-based grounding procedure to solve the task. In the next decision cycle, Voyager reasons over the environmental feedback to determine task completion. If successful, Voyager selects a learning action adding the grounding procedure to procedural memory; otherwise, it uses reasoning to refine the code and re-executes it. The importance of long-term memory and procedural learning is empirically verified by comparing to baselines like ReAct and AutoGPT and ablations without the procedural memory. Voyager is shown to better explore areas, master the tech tree, and zero-shot generalize to unseen tasks.

**Generative Agents** (Park et al., 2023) are language agents grounded to a sandbox game affording interaction with the environment and other agents. Its action space also has all four kinds of actions: grounding, reasoning, retrieval, and learning. Each agent has a long-term episodic memory that stores events in a list. These agents use retrieval and reasoning to generate reflections on their episodic memory (e.g., "I like to ski now.") which are then written to long-term semantic memory. During decision-making, it retrieves relevant reflections from semantic memory, then reasons to make a high-level plan of the day. While executing the plan, the agent receives a stream of grounding observations; it can reason over these to maintain or adjust the plan.

**Tree of Thoughts (ToT)** (Yao et al., 2023) can be seen as a special kind of language agent with only one external action: submitting a final solution to a reasoning problem (game of 24, creative writing, crosswords puzzle). It has no long-term memory, and only reasoning in its internal action space, but differs from all previous agents in its deliberate decision-making. During planning, ToT iteratively proposes, evaluates, and selects "thoughts" (reasoning actions) based on LLM reasoning, and maintains them via a tree search algorithm to enable global exploration as well as local backtrack and foresight.

# 6 Actionable Insights

Compared to some recent empirical surveys around language agents (Mialon et al., 2023; Weng, 2023; Wang et al., 2023b), CoALA offers a theoretical framework grounded in the well-established research of cognitive architectures. This leads to a unique and complementary set of actionable insights.

**Modular agents: thinking beyond monoliths.** Perhaps our most important suggestion is that *agents should be structured and modular.* Practically, just as standardized software is used across robotics platforms (Quigley, 2009; Macenski et al., 2022), a framework for language agents would consolidate technical investment and improve compatibility.

- **In academic research**, standardized terms allow conceptual comparisons across works (Table 2), and open-source implementations would further facilitate modular plug-and-play and re-use. For example, the theoretical framework of Markov Decision Processes (Puterman, 2014) provides a standardized set of concepts and terminology (e.g., state, action, reward, transition) for reinforcement learning (Sutton and Barto, 2018). Correspondingly, empirical frameworks like OpenAI Gym (Brockman et al., 2016) provided standardized abstractions (e.g., `obs, reward, done, info = env.step(action)`) that facilitate empirical RL work. Thus, it would be timely and impactful to also implement useful abstractions (e.g., `Memory`, `Action`, `Agent` classes) for language agents, and cast simpler agents into such an empirical CoALA framework as examples for building more complex agents.

- **In industry applications**, maintaining a single company-wide "language agent library" would reduce technical debt (Sculley et al., 2014; Lwakatare et al., 2020) by facilitating testing and component re-use across individual agent deployments. It could also standardize the customer experience: rather than interacting with a hodgepodge of language agents developed by individual teams, end users would experience a context-specific instantiation of the same base agent.

- **LLMs vs. code in agent design**. CoALA agents possess two forms of procedural memory: agent code (deterministic rules) and LLM parameters (a large, stochastic production system). Agent code is interpretable and extensible, but often brittle in face of stochasticity and limited to address situations the designer anticipates. In contrast, LLM parameters are hard to interpret, but offer significant zero-shot flexibility in new contexts (Huang et al., 2022b). CoALA thus suggests using code sparingly to implement generic algorithms that complement LLM limitations, e.g., implementing tree search to mitigate myopia induced by autoregressive generation (Yao et al., 2023; Hao et al., 2023).

**Agent design: thinking beyond simple reasoning.** CoALA defines agents over three distinct concepts: (i) internal memory, (ii) a set of possible internal and external actions, and (iii) a decision making procedure over those actions. Using CoALA to develop an application-specific agent consists of specifying implementations for each of these components in turn. We assume that the agent's environment and external action space are given, and show how CoALA can be used to determine an appropriate high-level architecture. For example, we can imagine designing a personalized retail assistant (Yao et al., 2022a) that helps users find relevant items based on their queries and purchasing history. In this case, the external actions would consist of dialogue or returning search results to the user.

- **Determine what memory modules are necessary.** In our retail assistant example, it would be helpful for the agent to have semantic memory containing the set of items for sale, as well as episodic memory about each customer's previous purchases and interactions. It will need procedural memory defining functions to query these datastores, as well as working memory to track the dialogue state.

- **Define the agent's internal action space.** This consists primarily of defining read and write access to each of the agent's memory modules. In our example, the agent should have read and write access to episodic memory (so it can store new interactions with customers), but read-only access to semantic and procedural memory (since it should not update the inventory or its own code).

- **Define the decision-making procedure.** This step specifies how reasoning and retrieval actions are taken in order to choose an external or learning action. In general, this requires a tradeoff between performance and generalization: more complex procedures can better fit to a particular problem (e.g., Voyager (Wang et al., 2023a) for Minecraft) while simpler ones are more domain-agnostic and generalizable (e.g., ReAct (Yao et al., 2022b)). For our retail assistant, we may want to encourage retrieval of episodic memory of interactions with a user to provide a prior over their search intent, as well as an explicit evaluation step reasoning about whether a particular set of search results will satisfy that intent. We can simplify the decision procedure by deferring learning to the end of the interaction (Shinn et al., 2023; Park et al., 2023), summarizing the episode prior to storing it in episodic memory.

**Structured reasoning: thinking beyond prompt engineering.** Early work on prompt engineering manipulated the LLM's input and output via low-level string operations. CoALA suggests a more structured reasoning procedure to update working memory variables.

- **Prompting frameworks** like LangChain (LangChain, 2022) and LlamaIndex (LlamaIndex, 2023) can be used to define higher-level sequences of reasoning steps, reducing the burden of reasoning per LLM call and the low-level prompt crafting efforts. **Structural output parsing solutions** such as Guidance (Guidance, 2023) and OpenAI function calling (OpenAI, 2023b) can help update working memory variables. Defining and building good working memory modules will also be an important direction of future research. Such modules may be especially important for industry solutions where LLM reasoning needs to seamlessly integrate with large-scale code infrastructure.

- **Reasoning usecases in agents can inform and reshape LLM training** in terms of the types (e.g., reasoning for self-evaluation, reflection, action generation, etc.) and formats (e.g., CoT (Wei et al., 2022b), ReAct (Yao et al., 2022b), Reflexion (Shinn et al., 2023)) of training instances. By default, existing LLMs are trained and optimized for NLP tasks, but agent applications have explored new modes of LLM reasoning (e.g., self-evaluation) that have proven broadly useful. LLMs trained or finetuned towards these capabilities will more likely be the backbones of future agents.

**Long-term memory: thinking beyond retrieval augmentation**. While traditional retrieval-augmented language models (Guu et al., 2020; Lewis et al., 2020; Borgeaud et al., 2022) only read from human-written corpora, memory-augmented language agents can both read and write self-generated content autonomously. This opens up numerous possibilities for efficient lifelong learning.

- **Combining existing human knowledge with new experience and skills** can help agents bootstrap to learn efficiently. For example, a code-writing agent could be endowed with semantic programming knowledge in the form of manuals or textbooks. It could then generate its own episodic knowledge from experience; reflect on these experiences to generate new semantic knowledge; and gradually create procedural knowledge in the form of a code library storing useful methods.

- **Integrating retrieval and reasoning** can help to better ground planning. Recent computational psychological models implicate an integrated process of memory recall and decision-making (Zhou et al., 2023a; Zhao et al., 2022) – suggesting that adaptive mechanisms interleaving memory search and forward simulation will allow agents to make the most of their knowledge.

**Learning: thinking beyond in-context learning or finetuning**. CoALA's definition of "learning" encompasses these methods, but extends further to storing new experience or knowledge, or writing new agent code (Section 4.5). Important future directions include:

- **Meta-learning by modifying agent code** would allow agents to learn more effectively. For example, learning better retrieval procedures could enable agents to make better use of their experience. Recent expansion-based techniques (Nogueira et al., 2019; Wang et al., 2023c; Tang et al., 2023a) could allow agents to reason about when certain knowledge would be useful, and store this as metadata to facilitate later recall. These forms of meta-learning would enable agents to go beyond human-written code, yet are understudied due to their difficulty and risk.

- **New forms of learning (and unlearning)** could include fine-tuning smaller models for specific reasoning sub-tasks (Zelikman et al., 2022; Huang et al., 2022a; Ahn et al., 2022), deleting unneeded memory items for "unlearning" (Nguyen et al., 2022c), and studying the interaction effects between multiple forms of learning (Tuyls et al., 2022; Park et al., 2023; Xie et al., 2023; Khattab et al., 2022).

**Action space: thinking beyond external tools or actions**. Although "action space" is a standard term in reinforcement learning, it has been used sparingly with language agents. CoALA argues for defining a clear and task-suitable action space with both internal (reasoning, retrieval, learning) and external (grounding) actions, which will help systematize and inform the agent design.

- **Size of the action space.** More capable agents (e.g., Voyager, Generative Agents) have larger action spaces – which in turn means they face a more complex decision-making problem. As a result, these agents rely on more customized or hand-crafted decision procedures. The tradeoff of the action space vs. decision-making complexities is a basic problem to be considered before agent development, and taking the minimal action space necessary to solve a given task might be preferred.

- **Safety of the action space.** Some parts of the action space are inherently riskier. "Learning" actions (especially procedural deletion and modification) could cause internal harm, while "grounding" actions (e.g., "rm" in bash terminal, harmful speech in human dialog, holding a knife in physical environments) could cause external harm. Today, safety measures are typically task-specific heuristics

(e.g., remove "os" operations in Python (Chen et al., 2021), filter keywords in dialog (Chowdhery et al., 2022; Driess et al., 2023), limit robots to controlled environments (Ahn et al., 2022)). However, as agents are grounded to more complex environments with richer internal mechanisms, it may be necessary to specify and ablate the agent's action space for worst-case scenario prediction and prevention (Yao and Narasimhan, 2023).

**Decision making: thinking beyond action generation**. We believe one of the most exciting future directions for language agents is decision-making: as detailed in Section 4.6, most works are still confined to proposing (or directly generating) a single action. Present agents have just scratched the surface of more deliberate, propose-evaluate-select decision-making procedures.

- **Mixing language-based reasoning and code-based planning** may offer the best of both worlds. Existing approaches either plan directly in natural language (Huang et al., 2022c; Ahn et al., 2022) or use LLMs to translate from natural language to structured world models (Wong et al., 2023; Liu et al., 2023a; Zhang et al., 2023a; Li et al., 2023a; Guan et al., 2023; Silver et al., 2022; 2023). Future work could integrate these: just as Soar incorporates a simulator for physical reasoning (Laird, 2022), agents may write and execute simulation code on the fly to evaluate the consequences of plans. See Section 7 for more discussion.

- **Extending deliberative reasoning to real-world settings**. Initial works have implemented classical planning and tree search (Yao et al., 2023; Hao et al., 2023; Liu et al., 2023a; Dagan et al., 2023), using toy tasks such as game of 24 or block building. Extending these schemes to more complicated tasks with grounding (Qin et al., 2023) and long-term memory is an exciting direction.

- **Metareasoning to improve efficiency.** LLM calls are both slow and computationally intensive. Using LLMs for decision-making entails a balance between their computational cost and the utility of the resulting improved plan. Most LLM reasoning methods fix a search budget by specifying a depth of reasoning (Yao et al., 2023), but humans appear to adaptively allocate computation (Russek et al., 2022; Lieder and Griffiths, 2020; Callaway et al., 2022; Gershman et al., 2015). Future work should develop mechanisms to estimate the utility of planning (Laidlaw et al., 2023) and modify the decision procedure accordingly, either via amortization (fine-tuning the LLM based on the results of previous actions, e.g. Nguyen, 2023; Hamrick et al., 2019), routing among several decision sub-procedures (e.g., ReAct (Yao et al., 2022b) investigated backing off to CoT (Wei et al., 2022b) and vice versa), or updates to the decision-making procedure.

- **Calibration and alignment.** More complex decision-making is currently bottlenecked by issues such as over-confidence and miscalibration (Jiang et al., 2021; Braverman et al., 2020; Chen et al., 2022), misalignment with human values or bias (Liang et al., 2021; Feng et al., 2023), hallucinations in self-evaluation (Shinn et al., 2023), and lack of human-in-the-loop mechanisms in face of uncertainties (Nguyen et al., 2022a; Ren et al., 2023). Solving these issues will significantly improve LLMs' utilities as agent backbones.

## 7  Discussion

In addition to the practical insights presented above, CoALA raises a number of open conceptual questions. We briefly highlight the most interesting as important directions for future research and debate.

**LLMs vs VLMs: should reasoning be language-only or multimodal?** Most language agents use language-only models for decision-making (Yao et al., 2022b; Wang et al., 2023a; Yao et al., 2023), employing a separate captioning model to convert environment observations to text when necessary (Ahn et al., 2022; Zeng et al., 2022). However, the latest generation of language models are multimodal, allowing interleaved image and text input (OpenAI, 2023a; Alayrac et al., 2022; Team et al., 2023; Li et al., 2023b). Language agents built on such multimodal models natively reason over both image and text input (Bavishi et al., 2023; Elsen et al., 2023; Liu et al., 2023b; Hong et al., 2023b; Driess et al., 2023), allowing them to ingest perceptual

data and directly produce actions. This bypasses the lossy image-to-text conversion; however, it also tightly couples the reasoning and planning process to the model's input modalities.

At a high level, the two approaches can be seen as different tokenization schemes to convert non-linguistic modalities into the core reasoning model's language domain. The modular approach uses a separate image-to-text model to convert perceptual data into language (Ahn et al., 2022; Zeng et al., 2022), while the integrated approach projects images directly into the language model's representation space (Bavishi et al., 2023; Elsen et al., 2023; Liu et al., 2023b). Integrated, multimodal reasoning may allow for more human-like behaviors: a VLM-based agent could "see" a webpage, whereas a LLM-based agent would more likely be given raw HTML. However, coupling the agent's perception and reasoning systems makes the agent more domain-specific and difficult to update. In either case, the basic architectural principles described by CoALA — internal memories, a structured action space, and generalized decision-making — can be used to guide agent design.

**Internal vs. external: what is the boundary between an agent and its environment?** While humans or robots are clearly distinct from their embodied environment, digital language agents have less clear boundaries. For example, is a Wikipedia database an internal semantic memory or an external digital environment (Yao et al., 2022b)? If an agent iteratively executes and improves code before submitting an answer (Shinn et al., 2023; Yang et al., 2023), is the code execution internal or external? If a method consists of proposal and evaluation prompts (Yao et al., 2023), should it be considered a single agent or two collaborating simpler agents (proposer and evaluator)?

We suggest the boundary question can be answered in terms of *controllability* and *coupling*. For example, Wikipedia is not *controllable*: it is an external environment that may be unexpectedly modified by other users. However, an offline version that only the agent may write to *is* controllable, and thus can be considered an internal memory. Similarly, code execution on a internal virtual environment should be considered an internal reasoning action, whereas code execution on an external machine (which may possess security vulnerabilities) should be considered an external grounding action. Lastly, if aspects of the agent – such as proposal and evaluation prompts – are designed for and dependent on each other, then they are *tightly coupled* and best conceptualized as components in an individual agent. In contrast, if the steps are independently useful, a multi-agent perspective may be more appropriate. While these dilemmas are primarily conceptual, such understanding can support agent design and help the field align on shared terminology. Practioners may also just choose their preferred framing, as long as it is consistent and useful for their own work.

**Physical vs. digital: what differences beget attention?** While animals only live once in the physical world, digital environments (e.g., the Internet) often allow sequential (via resets) and parallel trials. This means digital agents can more boldly explore (e.g., open a million webpages) and self-clone for parallel task solving (e.g., a million web agents try different web paths), which may result in decision-making procedures different from current ones inspired by human cognition (Griffiths, 2020).

**Learning vs. acting: how should agents continuously and autonomously learn?** In the CoALA framework, learning is a result action of a decision-making cycle just like grounding: the agent deliberately chooses to commit information to long-term memory. This is in contrast to most agents, which simply fix a learning schedule and only use decison making for external actions. Biological agents, however, do not have this luxury: they must balance learning against external actions in their lifetime, choosing when and what to learn (Mattar and Daw, 2018). More flexible language agents (Wang et al., 2023a; Park et al., 2023) would follow a similar design and treat learning on par with external actions. Learning could be proposed as a possible action during regular decision-making, allowing the agent to "defer" it until the appropriate time.

**GPT-4 vs GPT-N: how would agent design change with more powerful LLMs?** Agent design is a moving target as new LLM capabilities emerge with scale (Wei et al., 2022a). For example, earlier language models such as GPT-2 (Radford et al., 2019) would not support LLM agents — indeed, work at that time needed to combine GPT-2 with reinforcement learning for action generation (Yao et al., 2020); GPT-3 (Brown et al., 2020) unlocked flexible few-shot and zero-shot reasoning for NLP tasks; while only GPT-4 (OpenAI, 2023a) starts to afford more reliable self-evaluation (Saunders et al., 2022; Shinn et al., 2023; Yao et al., 2023) and self-refinement (Madaan et al., 2023; Chen et al., 2023b). Will future LLMs further reduce the need for coded rules and extra-learned models? Will this necessitate changes to the CoALA framework? As a thought experiment, imagine GPT-N could "simulate" memory, grounding, learning, and

decision-making in context: list all the possible actions, simulate and evaluate each one, and maintain its entire long-term memory explicitly in a very long context. Or even more boldly: perhaps GPT-N+1 succeeds at generating the next action by simulating these implicitly in neurons, without any intermediate reasoning in context. While these extreme cases seem unlikely in the immediate future, incremental improvements may alter the importance of different CoALA components. For example, a longer context window could reduce the importance of long-term memory, while more powerful reasoning for internal evaluation and simulation could allow longer-horizon planning. In general, LLMs are not subject to biological limitations (Griffiths, 2020), and their emergent properties have been difficult to predict. Nonetheless, CoALA – and cognitive science more generally – may still help organize tasks where language agents succeed or fail, and suggest code-based procedures to complement a given LLM on a given task. Even in the most extreme case, where GPT implements all of CoALA's mechanisms in neurons, it may be helpful to leverage CoALA as a conceptual guide to discover and interpret those implicit circuits. Of course, as discussed in Section 6, agent usecases will also help discover, define and shape LLM capabilities. Similar to how chips and computer architectures have co-evolved, language model and agent design should also develop a reciprocal path forward.

## 8    Conclusion

We proposed Cognitive Architectures for Language Agents (CoALA), a conceptual framework to describe and build language agents. Our framework draws inspiration from the rich history of symbolic artificial intelligence and cognitive science, connecting decades-old insights to frontier research on large language models. We believe this approach provides a path towards developing more general and more human-like artificial intelligence.

## Acknowledgements

We thank Harrison Chase, Baian Chen, Khanh Nguyen, Ofir Press, Noah Shinn, Jens Tuyls for proofreading and valuable feedback, and members from the Princeton NLP Group and Princeton Computational Cognitive Science Lab for helpful discussions. Finally, we thank our anonymous reviewers for insightful comments and suggestions. SY and KN acknowledge support from an Oracle Collaborative Research award and the National Science Foundation under Grant No. 2239363. Any opinions, findings, conclusions, or recommendations expressed in this material are those of the author(s) and do not necessarily reflect the views of the National Science Foundation. SY is also supported by the Harold W. Dodds Fellowship from Princeton. TS is supported by the National Defense Science and Engineering (NDSEG) Graduate Fellowship Program.

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
