# OpenReview forum: "Cognitive Architectures for Language Agents"
_TMLR — Accepted by TMLR_

### Review · Reviewer_EJcm · 2023-12-06

**Summary Of Contributions:**

This submission provides two main contributions.  The first is a framework for agents that utilize an LLM as key internal component.  The authors call this framework **Cognitive Architectures for Language Agents (CoALA)**.  This framework incorporates ideas from previous cognitive architecture research, with some modifications for the new setting.  The second main contribution of the submission is a categorization of the work that has been abundant in recent years in this area: LLM-based agents for many settings.  The CoALA framework provides a way to discuss and contextualize the approaches that the various researchers have taken.  These two main contributions lead naturally to one additional final contribution, an identification of possible fruitful or underexplored areas for future work.

**Audience:**

Yes

**Broader Impact Concerns:**

I don't have any broader impact concerns that would necessitate adding a Broader impact statement.

**Claims And Evidence:**

Yes

**Requested Changes:**

I have no changes to request.

**Strengths And Weaknesses:**

Strengths of the submission:
- I found the submission very well-written.  It was easy to read, understand, and follow.
- I am personally encouraged by the perspective of this paper. It identifies and utilizes contributions from earlier lines of research (in this case production systems), and shows how they provide valuable insight and direction, even in the face of the novel advancements being made.  This perspective is important, and in this case the connection is clearly described and shown convincingly to be useful.
- This submission is very timely.  The recent (and ongoing) breakthroughs with LLMs have led to a rapid "wild west" where researchers (and business/lay people) from almost every area imaginable are exploring ways to utilize the amazing abilities of LLMs to make better agents for other problems.  I think that a conceptual framework like this comes along at a good time where it can help steer this energy towards making progress over time and more systematically understanding at least one set of ways to think about these types of agents and how they can be designed and implemented.
- The organization of the submission was very good.  Each question I had as I read through, (ie I wonder what some complete previous agents look like wrt to CoALA, or what does this mean for future research directions) was met by a section or section that specifically addressed that topic or issue.  I thought specifically the **Case Studies**, **Actionable Insights** , and **Discussion** sections were each specifically well-designed and help the paper be strong.
- The breadth and pervasiveness of the references helps convey a strong sense of the state of the art, and the many, many things that have been done in this area. In other words, the excitement and activity in this area comes across strongly.

Weaknesses
- The only thing that I noticed was that the beginning of the **Discussion** section was abrupt.  This is more a personal thing; I am not a big fan of subsection headings immediately following section headings, without some sort of introductory or connecting text.

---

> ### Author Response · Authors · 2024-01-03
> **Thank you!**
>
> > I found the submission very well-written. It was easy to read, understand, and follow. I am personally encouraged by the perspective of this paper... in this case the connection is clearly described and shown convincingly to be useful. This submission is very timely. … The organization of the submission was very good. …The breadth and pervasiveness of the references helps convey a strong sense of the state of the art, and the many, many things that have been done in this area. In other words, the excitement and activity in this area comes across strongly.
>
> Thank you so much for the kind words – we’re glad you appreciate the contribution and share your hope it will encourage further research in this area!
>
> ---
> Concern:
> > The only thing that I noticed was that the beginning of the Discussion section was abrupt. This is more a personal thing; I am not a big fan of subsection headings immediately following section headings, without some sort of introductory or connecting text.
>
> Thanks for the suggestion! On re-reading the manuscript we agreed the Discussion begins abruptly. We’ve added the following text at the beginning of the Discussion to distinguish between Sections 6 and 7 (p. 17 in the updated paper):
>
> *In addition to the practical insights presented above, CoALA raises a number of open conceptual questions. We briefly highlight the most interesting as important directions for future research and debate.*

---

### Review · Reviewer_W51p · 2023-12-21

**Summary Of Contributions:**

The paper proposes a theoretical framework grounded in cognitive architectures and uses it to organize the language agents literature.

**Audience:**

Yes

**Broader Impact Concerns:**

None.

**Claims And Evidence:**

Yes

**Requested Changes:**

In my view, to strengthen the work, a more in depth discussion about multimodal LLMs and why they can fit or not into the CoALA framework would be useful.

Small typo:
page 15: an unnecessary comma before CoT.

**Strengths And Weaknesses:**

Strengths:
- Efforts to make the framework general enough.
- Comprehensive view of the related literature.

Weaknesses:
- The multimodality is not discussed enough. In the framework every modalities (e.g. images) have to be converted to text. This restriction excludes some models that were explicitly trained to be multimodal (with tokens from different modalities).

---

> ### Author Response · Authors · 2024-01-03
> **On multimodality**
>
> > Strengths: Efforts to make the framework general enough.
> Comprehensive view of the related literature.
>
> Thank you – we’re glad you appreciate the comprehensive + general framing of language agents!
>
> ---
>
> Concern:
> > The multimodality is not discussed enough. In the framework every modalities (e.g. images) have to be converted to text. This restriction excludes some models that were explicitly trained to be multimodal (with tokens from different modalities).
>
> Requested change:
> > In my view, to strengthen the work, a more in depth discussion about multimodal LLMs and why they can fit or not into the CoALA framework would be useful.
>
> Thank you for suggesting this! We briefly mentioned multimodality in Section 4.2 but agree this is an important trend which is worth more discussion. We have added a new section to the Discussion to cover multimodal LMs (in blue in the revised paper), which we have copied below for convenience. Please let us know if you have further comments or suggestions in this area.
>
> ***LLMs vs VLMs: should reasoning be language-only or multimodal?** Most language agents use language-only models for decision-making (Yao et al., 2022b; Wang et al., 2023a; Yao et al., 2023), employing a separate captioning model to convert environment observations to text when necessary (Ahn et al., 2022; Zeng et al., 2022).  However, the latest generation of language models are multimodal, allowing interleaved image and text input (OpenAI, 2023a; Alayrac et al., 2022; Team et al., 2023; Li et al., 2023b). Language agents built on such multimodal models may natively reason over both image and text input (Bavishi et al., 2023; Elsen et al., 2023; Liu et al., 2023b; Hong et al., 2023b; Driess et al., 2023), allowing them to ingest perceptual data and directly produce actions. This approach bypasses the lossy image-to-text conversion; however, it also tightly couples the reasoning and planning process to the model’s input modalities.*
>
> *At a high level, the two approaches can be seen as different tokenization schemes to convert non-linguistic modalities into the core reasoning model’s language domain. The modular approach uses a separate image-to-text model to convert perceptual data into language (Ahn et al., 2022; Zeng et al., 2022), while the integrated approach projects images directly into the language model’s representation space (Bavishi et al., 2023; Elsen et al., 2023; Liu et al., 2023b). Integrated, multimodal reasoning may allow for more human-like behaviors: a VLM-based agent could “see” a webpage, whereas a LLM-based agent would more likely be given raw HTML. However, coupling the agent's perception and reasoning systems makes the agent more domain-specific and difficult to update. In either case, the basic architectural principles described by CoALA — internal memories, a structured action space, and generalized decision-making — can be used to guide agent design.*
>
> ---
>
> > Small typo: page 15: an unnecessary comma before CoT.
>
> Thank you for flagging! To the best of our knowledge it is correct American English style to include a comma after “e.g.” so we will keep the paper as-is.

---

### Review · Reviewer_TjWi · 2023-12-21

**Summary Of Contributions:**

This paper introduces a new conceptual framework, named CoALA, aimed at helping to categorize existing language agents and finding insights on what grounds are left still to be explored. Authors present examples of how existing work can be categorized by CoALA.

**Audience:**

Yes

**Broader Impact Concerns:**

This work proposes a conceptual framework for general capable agents, which might have potential misuse concerns. Having say that, this work is only a positional paper highlighting paths for the field to follow. I believe no harmful applications or special concerns can be directly obtained from this work.

**Claims And Evidence:**

Yes

**Requested Changes:**

Changes critical for securing recommendation:

* Making either a clearer outlining of the contributions and the aim of CoALA in the lines of my first concern above.

* Tackling the problem CoALA helps at systematically building and designing agents. At its current state it is clear that CoALA helps in that end but not is that it is systematic and if so, how it can be procedurally used by  the field. I would suggest either removing the word systematic from the claim or rather I would include a pseudoalgorithm / rule chain / systematic example on how to both understand  existing solutions and building new ones. Current examples for instance explain how  some existing works are labelled according to CoALA but not how the "systematic categorization" goes.

Changes that would strength the work:

* Adding connections at the intros of the different sections in the background on how they play a role in CoALA
* Since this work is presenting a connection framework leverages strengths from production systems and cognitive architectures it would be good to mention [1] that also merges those two lines of work for a different objective.

[1] Xia, Yuchen, et al. "Towards autonomous system: flexible modular production system enhanced with large language model agents."


---Post rebuttal---
Authors updated their claims, clarified the objectives and added explanations and examples that addressed my concerns and make this work better positioned and easier to use for future readers, I am updating my evaluation of claims accordingly

**Strengths And Weaknesses:**

The emergence of a myriad of works on complex systems for LLMs agents make a work like this one aimed at producing a systematic classification of such systems a needed piece of work. Authors also do a great efforts on presenting the history context of AI and language agents and contextualizing the current state of the field.

However, I have concerns about the current state of the paper regarding focus, presentation and clarity, which are of special relevance given that this is exclusively a positioning paper and that no empirical evaluation is present. Specifically:

* In the intro CoALA seems to be aimed at categorizing existing work and proposing new directions of missing research. However, as the reader progresses it seems that the authors are presenting this as a novel conceptual framework to design general purpose AI agents that also helps at categorizing existing research and finding unexplored ground in the field. Either intro
* There is a large body of background in this work that would benefit of clearer explanations/connections/intros on how their pieces are specifically important for understanding CoALA
*The work fails at presenting clearly the method proposed to systematically categorize and build existing agents.

---

> ### Author Response · Authors · 2024-01-03
> **Clarifying claims and background**
>
> > The emergence of a myriad of works on complex systems for LLMs agents make a work like this one aimed at producing a systematic classification of such systems a needed piece of work. Authors also do a great efforts on presenting the history context of AI and language agents and contextualizing the current state of the field.
>
> We’re glad you appreciate the timeliness and importance of our framework, as well as our effort to place recent work within the broader historical context!
>
> ---
>
> Concern:
> > In the intro CoALA seems to be aimed at categorizing existing work and proposing new directions of missing research. However, as the reader progresses it seems that the authors are presenting this as a novel conceptual framework to design general purpose AI agents that also helps at categorizing existing research and finding unexplored ground in the field….
>
> Requested change:
> > Making either a clearer outlining of the contributions and the aim of CoALA in the lines of my first concern above.
>
> We apologize if the introduction felt mismatched to the contribution of the paper. We are indeed intending CoALA as a conceptual framework to both characterize and design agents. We have updated the introduction accordingly (p. 2 in the revised paper):
>
> *“Thus, we propose Cognitive Architectures for Language Agents (CoALA), a conceptual framework to characterize and design general purpose language agents.”*
>
> Please let us know if you have further concerns here!
>
>
> ---
>
> Concern:
> > The work fails at presenting clearly the method proposed to systematically categorize and build existing agents.
>
> Requested change:
> > Tackling the problem CoALA helps at systematically building and designing agents. At its current state it is clear that CoALA helps in that end but not is that it is systematic and if so, how it can be procedurally used by the field. I would suggest either removing the word systematic from the claim or rather I would include a pseudoalgorithm / rule chain / systematic example on how to both understand existing solutions and building new ones. Current examples for instance explain how some existing works are labelled according to CoALA but not how the "systematic categorization" goes.
>
> To clarify, our contribution is a “blueprint” for general-purpose language agents, rather than a “procedure” to specify the correct architecture for a particular application. We have removed the word “systematic” from the abstract to avoid giving that impression.
>
> That said, we recognize that it is important to help readers use CoALA in practice. We have added a new subsection to Section 6 (*“Agent design: choosing the right architecture”*) which provides high-level guidance and an example demonstrating how to apply CoALA to design a new agent.
>
> Please let us know if this helps address your concern — we’d be happy to provide further information or tone down parts you find overclaiming.
>
> ---
>
> Concern:
> > There is a large body of background in this work that would benefit of clearer explanations/connections/intros on how their pieces are specifically important for understanding CoALA
>
> Requested change:
> > Adding connections at the intros of the different sections in the background on how they play a role in CoALA
>
> Thank you for suggesting this! The background is intended to provide historical context (Section 2) and explain why cognitive architectures are an appropriate source of inspiration for designing language agents (Section 3). The original paper noted that these sections are not essential to understand CoALA (p. 2), but are important to motivate why design principles from cognitive architectures can help understand and design language agents.
>
> We have updated the Introduction, Section 2 (Background) and Section 3 (Connections between LMs and Production Systems) to further clarify how these support the core ideas in the CoALA framework (Section 4). Please see the revised paper, with changes highlighted in blue.
>
> ---
>
> Requested change:
> > Since this work is presenting a connection framework leverages strengths from production systems and cognitive architectures it would be good to mention [1] that also merges those two lines of work for a different objective.
>
> > [1] Xia, Yuchen, et al. "Towards autonomous system: flexible modular production system enhanced with large language model agents."
>
> Thank you for bringing this paper to our attention. We note that Xia et. al. address a different kind of production system (specifically, manufacturing and industrial automation, rather than the logical production systems we consider). However, we agree that it is an excellent example of language agents applied to a new domain, and have added it to the paper in our background section on language agents (Section 2.4).

---

> > ### Comment · Reviewer_TjWi · 2024-01-04
> > **Response to Authors**
> >
> > I want to thank the authors for their responses and clarifications. My concerns have been well addressed in the new revision and I will be updating my review accordingly

---

### Comment · Editors_In_Chief · 2025-12-02

Congratulations to the authors on this paper being named a 2025 Outstanding Certification Finalist! It has also been awarded a Featured Certification in recognition of this.

For more information, see https://medium.com/@TmlrOrg/announcing-the-2025-tmlr-outstanding-certification-e26d548ff011.

---

### Decision · Action_Editor_TemM · 2024-02-03

**Recommendation:** Accept as is

**Comment:**

This is a positional paper with thorough literature survey and clear writings. All reviewers found the paper to be of value to TMLR community and recommended acceptance. Given that the recent LLM agent research is frequently driven by empirical practices, such a conceptual paper is of unique value to the community, and therefore I recommend acceptance. One recommendation is to use the breakdowns in Section 6 to categorize and describe the existing LLM agent work in a table. This could help validate that this new conceptual framework could successfully offer directions of improvements.

**Audience:**

This paper should be relevant to many researchers and engineers working on LLM and especially LLM agents. It helps contextualize the recent surge of LLM agent research in the rich history of classic symbolic AI and cognitive science research.

A potential counterargument is that given how fast empirical developments in this space are proceeding, practitioners may offer more insights than theoretical conceptual framings.

**Claims And Evidence:**

Claim: propose Cognitive Architectures for Language Agents (CoALA), a framework to organize existing LLM agents and plan future
developments
- accurate and convincing to all reviewers.

Claim: use CoALA to retrospectively survey and organize a large body of recent work, and prospectively identify actionable directions towards more capable agents
- authors do bridge a rich history of work on symbolic reasoning and AI research with recent LLM-based empirical work, and offer a list of actionable items at the end of the paper